# Treatment of Parkinson’s Disease through Personalized Medicine and Induced Pluripotent Stem Cells

**DOI:** 10.3390/cells8010026

**Published:** 2019-01-07

**Authors:** Theo Stoddard-Bennett, Renee Reijo Pera

**Affiliations:** 1Department of Cell Biology and Neurosciences, Montana State University, Bozeman, MT 59717, USA; renee.reijopera@montana.edu; 2Department of Chemistry and Biochemistry, Montana State University, Bozeman, MT 59717, USA

**Keywords:** induced pluripotent stem cells, Parkinson’s disease, alpha-synuclein, cell- and tissue-based therapy, disease modeling, dopaminergic neurons

## Abstract

Parkinson’s Disease (PD) is an intractable disease resulting in localized neurodegeneration of dopaminergic neurons of the substantia nigra pars compacta. Many current therapies of PD can only address the symptoms and not the underlying neurodegeneration of PD. To better understand the pathophysiological condition, researchers continue to seek models that mirror PD’s phenotypic manifestations as closely as possible. Recent advances in the field of cellular reprogramming and personalized medicine now allow for previously unattainable cell therapies and patient-specific modeling of PD using induced pluripotent stem cells (iPSCs). iPSCs can be selectively differentiated into a dopaminergic neuron fate naturally susceptible to neurodegeneration. In iPSC models, unlike other artificially-induced models, endogenous cellular machinery and transcriptional feedback are preserved, a fundamental step in accurately modeling this genetically complex disease. In addition to accurately modeling PD, iPSC lines can also be established with specific genetic risk factors to assess genetic sub-populations’ differing response to treatment. iPS cell lines can then be genetically corrected and subsequently transplanted back into the patient in hopes of re-establishing function. Current techniques focus on iPSCs because they are patient-specific, thereby reducing the risk of immune rejection. The year 2018 marked history as the year that the first human trial for PD iPSC transplantation began in Japan. This form of cell therapy has shown promising results in other model organisms and is currently one of our best options in slowing or even halting the progression of PD. Here, we examine the genetic contributions that have reshaped our understanding of PD, as well as the advantages and applications of iPSCs for modeling disease and personalized therapies.

## 1. Introduction

Neurodegenerative diseases continue to pose increasing physical and financial burdens in an aging world, despite centuries of study. These disorders include Parkinson’s disease (PD), Alzheimer’s disease (AD), amyotrophic lateral sclerosis (ALS), Batten disease, and Huntington’s disease, among others. Since their discovery, neurodegenerative diseases have been clinically identified through post-mortem and physical examination. Formation of protein aggregates, localized neuronal death, and progressive symptoms are all typical of neurodegeneration; however, our knowledge of the pathogenesis and etiology underlying these disorders has been rebuilt due to recent advances in the field of genetics. Spurred by regenerative medicine, research in personalized medicine continues to search for underlying pathological mechanisms and novel therapies to halt or even slow the progression of these crippling diseases using stem cells (Figure 1). With advances in cellular reprogramming, iPSCs provide the greatest potential for cell therapy of localized neurodegeneration as found in PD. Due to iPSCs patient-specificity, the likelihood of an immune response after transplantation can be significantly reduced.

## 2. Pathology of Parkinson’s Disease

PD is the second most common neurodegenerative disease, debilitating 1% of the population over the last 60 years [1]. As such, it poses a special problem in aging societies [2,3]. Though many neuronal networks are affected by PD, it is the dopaminergic neurons (DAn) of the substantia nigra pars compacta (SNpc) [4,5,6] and cholinergic nucleus Basalis of Meynert that are most acutely lost [7]. The localized cell death of the SNpc results in disruption of the basal ganglia’s motor control network, causing the characteristic motor symptoms of PD—bradykinesia, tremors, rigidity and other changes in speech and gait. PD’s histopathology is marked by the presence of Lewy bodies and Lewy neurites that contain misfolded alpha-synuclein [8]. Found within Lewy bodies and Lewy neurites, the expression of alpha-synuclein has been attributed to PD’s pathogenesis. The exact function of unaffected alpha-synuclein, thought to also assist in vesicle turnover and synaptic release, remains unknown [9]. However, one study suggests that alpha-synuclein, when properly folded into its tetramer, exhibits protective properties and slows Lewy body formation and aggregation [10]. Lewy body formation typically begins in the SNpc, but a progressive spread to other structures in the brain has also been documented [11]. The role of alpha-synuclein has increased attention towards other non-motor symptoms of PD which include autonomic dysfunction, sleep disorders and neuropsychiatric symptoms (depression, psychosis, hallucinations) [12]. Many etiological questions remain. Although motor symptoms are only documented in 80% of revealed post-mortem alpha-synucleinopathy [13], these motor symptoms persist as a hallmark of clinical diagnosis. Here we examine the discoveries leading to our current understanding of PD, as well as propose iPSC transplantation as one of the most viable forms of disease-modifying therapy for PD.

## 3. Major Genetic Discoveries in Parkinson’s Research

In 2018, at least 5 major autosomal dominant genes, 5 autosomal recessive or X-linked factors and 11 monogenetic mutations for other disorders that present with Parkinsonian-like symptoms have been identified [14]. The most notable of these are mutations and polymorphisms of Leucine-rich repeat kinase 2 (*LRRK2*), a gene that plays a role in neuronal survival [15], and the *SNCA* gene which encode for a protein called alpha-synuclein (Table 1) [16]. However, while strongly supported by a large body of statistical evidence [17], the effect of all known genetic mutations and risk-enhancing polymorphisms combined only explain a portion of the genetic risk of disease. The heterogeneity of genetic factors only serves to highlight the complex interplay in neurodegeneration. These mutations may not be causal; they can, however, elevate risk 2- to 3-fold [18]. Patient-specific cell lines and powerful gene-editing tools now allow the study of these mutations in isolation. Current advances in genetic probing will only allow for sharper analysis in genetic counseling, enhanced understanding of PD’s progression and ultimately patient-specific treatments.

In 1997, a novel, rare mutation was identified in the *SNCA* gene that coded for a relatively unknown protein called alpha-synuclein [16]. The missense mutation (A53T) resulted in autosomal dominant PD inheritance that could be tracked through the hereditary line with almost full penetrance. Additionally, five other missense mutations to the *SNCA* gene, *A30P*, *E46K*, *H50Q*, *G51D* and *G209A* have also been reported with varying ages of PD onset [14]. More common duplications and triplications of the *SNCA* gene were later linked in a family known as the Iowa Kindred. The double and triple doses resulted in overexpression of natural alpha-synuclein and pathological PD [19].

In 2002, Funayma et al. reported that a region of chromosome 12 was found to be linked to PD inheritance in a Japanese family [20,21]. Two years later, the gene of interest was identified as *LRRK2* [22]. Mutations to *LRRK2* are by far the most common cause of genetic influence on PD [21,23]. Many other mutations of *LRRK2* have been reported, but few remain statistically significant. Inheritance follows an autosomal dominant pattern with an age-related penetrance ranging from 28% at age 59 to 74% at 79 [24]. *LRRK2* mutations comprise 4% of reported familial PD, and most cases exhibit pathology indistinguishable from sporadic PD with both Lewy body formation and DAn death [22,24]. PD from *LRRK2* heredity follows the typical pattern with an onset later in life and excellent response to levodopa (L-Dopa), a precursor to dopamine that can pass the blood-brain barrier, whereas *SNCA* inheritance is earlier-onset. Curiously, patients with *LRRK2* PD experience less severe motor symptoms associated with the frequency of falls and progression of dyskinesia [24]. Studies in cellular models that harbor these mutations show increased kinase activity resulting in neuro-oxidative stress and toxicity [25,26]. Although the protein is multifunctional, *LRRK2* knock-downs inhibit differentiation from neural progenitors to DAns and increase cell death [15]. These findings suggest LRRK2’s facilitation in cell survival and differentiation in the ventral midbrain.

Genetic loci have also been identified in familial PD that follow autosomal recessive inheritance. Two genes, phosphate and tensin homolog-induced putative kinase 1 (*PINK1*) and Daisuke-Junko-1 (*DJ-1*), are of special interest because they are involved in neuronal survival under cellular stress (Table 1). In cases of homozygosity, both *PINK1* and *DJ-1* mutations result in very early onset in the 30’s, low response to L-Dopa treatment and slow disease progression [27,28]. Additionally, an astute clinical observation of the comorbidity between Gaucher disease (GD) and PD led researchers to examine other proteins with suspect. GD is an autosomal recessive disease resulting from homozygous mutations to housekeeping glucocerebrosidase gene (GBA) (Table 1). GBA, a lysosomal enzyme of the CNS, is thought to also have a role in protein aggregation in PD when mutated [29]. The symptoms and pathophysiology closely follow the progression of PD, even L-Dopa therapy is effective for treating *GBA* mutation. Post-mortem analysis also reveals Lewy body and Lewy neurite formation. *GBA* mutations do not natively aggregate as cleanly in PD inheritance and only display 29.7% penetrance in populations over the age of 80 [29]. *GBA* mutations have been proven in 2009 to be the most common genetic risk factor for PD so far—present in 3–7% of idiopathic PD cases [30]. Technological advances combined with ever-increasing sample sizes and collaboration will hopefully uncover more sources of genetic and epigenetic influence. A summary of the genes discussed above are summarized in Table 1.

## 4. Tailored Therapies of Genetic Parkinson’s Disease

The pathology behind PD involves the dysfunction of multiple systems and neurotransmitters that lead to similar symptoms. However, the highly variable and clustered patient responses to L-Dopa treatment suggests that different biochemical mechanisms of degeneration may necessitate differing treatment approaches [31,32]. These differences include variation in motor response, dyskinesias, psychotic episodes, sleep disturbances and visual hallucinations. The approach of personalized medicine offers more effective clinical therapies and added benefit for patients living with specific subsets of PD. Personalized medicine is widely conflated with precision medicine when discussing medical interventions and we will use both interchangeably to consider targeted therapies for a given subpopulation of PD patients, as well as holistically approaching their condition.

The influence of specific genetic factors could explain some of the variability in individual drug response. Genetic differences play a large role in the metabolic pathway of drug uptake which could affect response to treatments [33,34]. Though much of the heritability of PD has yet to be explained, the field of pharmacogenomics attempts to examine the effects of genetics on drug mechanics—specifically the interplay of dopamine transporters, receptors and enzymes critical for dopamine processing for PD. Patients with familial *LRRK2* PD could be treated with LRRK2 inhibitors and pharmaceutical companies have begun clinical trials earlier this year (ID: NCT03710707) [35,36]. Conversely, specific drug regimens carry added risk for specific genotypes. Those harboring a polymorphism of the dopamine active transporter 1 gene (*DAT1*), a gene involved in the reuptake of dopamine in chemical synapses, are 2.5 times more likely to develop dyskinesias when treated with L-Dopa while a different point mutation of *DAT1* was shown to increase the risk of hallucination when treated with dopaminergic drugs [34,37]. Genetic factors are also influenced by gender for treatment. Dopamine receptor D2 (DRD2), a dopaminergic receptor subtype, polymorphisms protect L-Dopa induced dyskinesias in men but do not has protective effects for women [38]. Genetic differences can also be influenced by epigenetic factors as well. In the future, clinicians may also personalize treatment modalities based on the physiological progression of the individual as opposed to the severity of symptoms. Researchers are developing alpha-synuclein tracers to examine the localization and severity of alpha-synuclein formation in vivo, however, this technology is not currently available [39,40,41]. Though the implementation of personalized medicine based on specific subtype offers many benefits for patients, treatment modalities may require constant modification due to the progressive and dynamic nature of PD.

Ultimately, personalized treatment of PD embodies the holistic span of age, lifestyle, genotype, personality and other concurrent illnesses. This strategy is relatively new and ideally rests on a multidisciplinary team to ensure all the patients’ specific needs are considered before pursuing a given treatment plan. Patients are assessed by disease sub-group but, above all, treatments should be tailored to the individual. Genotype is not the only factor in prescribing treatments. Ropinirole and other dopamine agonists work for some patients, but their increased risk of impulse control disorders warrants caution if the individual already has impulsive tendencies [42]. Because PD afflicts an aging population, bone health and fall risk should be weighed with motor treatments and supplemented with vitamin D2 and bisphosphonates. For patients with cardiovascular issues, quetiapine and other antipsychotics that prolong the QTc interval should be avoided. Similarly, those with vascular impairments in the brain must have their homocysteine levels regulated when undergoing high dose L-Dopa treatment. Endocrine issues may also affect treatment plans. Those with hyperthyroidism or low body weight may receive lower levels of L-Dopa treatment. With new information and conflicting studies, best-practice guidelines continue to be altered and so no definite methods of therapy can be provided [34]. This places the onus on researchers to provide more nuanced insight using model systems that better represent PD pathology in order to deliver better care.

## 5. Pluripotent Stem Cells

To better understand Parkinson’s disease, researchers have sought models that mirror PD’s phenotypic manifestations as closely as possible. To date, researchers have used model organisms (yeast, mice, Drosophila and non-human primates) in three ways. First, organisms were subjected to direct injections into the central nervous system of 6-hydroxydopamine (6-OHDA) and 1-methyl-4-phenyl-1,2,5,6-tetrahydropyridine (MPTP) to mimic DAn cell death in the SNpc through oxidative stress. While helpful in examining the effects of blocking dopamine expression, this artificially induced model fails to reconstruct the underlying neuropathology of highly-sensitive DAn and typical formation of Lewy neurites and Lewy bodies. Second, overexpression of the human risk factors, such as the *SNCA* gene, in mouse models has shown age-dependent DAn degeneration similar to the human pathological phenotype [8,43,44]. These results support a causal role for alpha-synuclein in PD progression but currently, lack clinical application. Finally, human embryonic stem cells (hESC) and induced pluripotent stem cells (iPSC) have been grafted into model organisms in an existing PD state. This procedure has shown mixed but promising results [45,46,47,48,49,50,51].

Both hESCs and iPSCs possess unique qualities that make them ideal candidates for studying the development of PD [52]. Stem cells can be tailored to differentiate into a host of cell fates, including DAn of the SNpc that model PD on a cellular level [53,54]. Induced stem cells are also patient-specific, opening a window to the individual contribution of mutation and polymorphism risk factors on PD in a phenotypically similar state. DAn longevity can also be compared with other iPSC neurons such as cortical and olfactory neurons to probe the hypersensitivity of DAns specifically in the SNpc. Furthermore, the advent of TALEN, CRIPSR and other genetic reprogramming technologies have been applied to patient lines of iPSCs and extensively reviewed [55,56]. Corrected mutation lines can then be examined. Deriving a high percentage of fully functional, mature DAn of the ventral midbrain can be quite challenging and costly to scale up. The technical, in addition to ethical, obstacles of iPSC treatment may limit the feasibility of transplanting reprogrammed stem cells, but this opportunity in PD treatment is unprecedented. The first human clinical trial to transplant DAns from an iPSC source begins this year in Japan (ID: UMIN000033564) [57].

## 6. iPSCs as Disease Model for Parkinson’s Disease

The contribution of pharmacogenomics has been heightened through the use of patient-specific iPSC lines and genetic engineering technology to manipulate them. Ever since Yamanaka’s discovery in 2007 that a handful of transcription factors can reprogram cellular differentiation, iPSCs have been utilized extensively in the study of neurodegenerative disease to direct patient-specific cell fate [58,59]. While still limited in scope, iPSCs are currently the most robust and phenotypically similar model for PD [60]. Mutations of consequence can now be captured in iPSC lines and directed by small molecules to a DAn fate in PD models—all within a dish. Displayed openly, the real-time cellular effects of mutation can be physically observed and studied in tandem with control lines to limit genetic background effects of the affected individual; similarly, effects of oxidative stress common to PD can also be quantified with broad clinical applications for drug screening without human side-effects. Not surprisingly, it remains difficult to physically confirm the mechanisms of neurodegeneration and neuroprotection implicated by iPSC research as patients’ neurons are hidden deep within the brain. These effects similarly cannot be perfectly translated into the cellular environment of PD due to some epigenetic effects of aging eliminated in reprogramming protocols.

Differentiation of iPSCs to midbrain DAns begins with the initial dedifferentiation. In the last 10 years, thousands of iPSC lines have been generated by overexpressing certain transcriptional factors in somatic cells to bring them back to a pluripotent state. Those methods have been extensively reviewed [55,56]. Current methods of reprogramming employ episomes, viruses and synthetic mRNA to upregulate expression of the transcription factors without genomic integration that leads to tumorigenesis [61,62,63]. Dedifferentiation often results in mutation and, consequently, not all iPSC lines are of equal quality. Though iPSC reprogramming theoretically results in a clonal copy of the genome, sequencing entire genomes of iPSCs have revealed an average of 6 de novo mutations during reprogramming in coding regions [64]. Extensive quality controls are in place to measure the quality of iPSC lines such as expression of neural markers, transcriptome analysis of iPSC lines with available data of hESC expression as well as bioinformatics tests like Pluritest [65]. These safeguards are critical if clinicians wish to implement iPSCs further in personalized medicine.

Differentiation from pluripotent stem cells to mature DAns of the midbrain mimics a specific pathway in embryological development. Initially dopaminergic neurons of the SNpc were originally thought to have derived from neuroepithelial cells like other cortical neurons, but in fact, they are similar to the spinal cord, derived from the ventral floor plate of the neural tube [66]. The embryologic origin was confirmed by expression of other floor-plate markers such as Lmx1a and FoxA2 [67,68]. Differentiation protocols using small molecules and neurotrophic factors mimic in vivo neural floor-plate patterning by activating the sonic hedgehog (SHH) pathway, inhibition of SMAD and addition of FGF8 [69,70]. Differentiation through transfecting transcriptional factors can also be achieved, but spontaneous integration prohibits these methods from any clinical-grade application. Tuning DAn cell type is achieved with the addition of the WNT signaling molecule CHIR99021 (CHIR). The more CHIR added, the more hindbrain characteristics DA neurons adopt [71]. With the correct amount, DAns of the SNpc can be achieved that express characteristic GIRK2 markers.

Neurogenesis of localized DAns in iPSC lines provides unparalleled modeling of human conditions in PD. In stem cell models, unlike other model organisms, endogenous cellular machinery and transcriptional feedback are preserved, a fundamental step in accurately modeling this genetically complex disease. iPSCs have also been used to model AD, suggesting broader applications to a whole range of neurodegenerative disorders [72]. Furthermore, iPSC lines can now be maintained with a natural susceptibility to PD pathology without unnaturally high oxidation from MPTP. Genetic effects may be further isolated by the implementation of CRISPR/Cas9 editing to reduce genetic and clonal variation. iPSC mutation models can be additionally genetically corrected at dedifferentiation and co-cultured with mutant lines to control for epigenetic and passage state [73]. Though reprogramming technologies have been used on patients with idiopathic PD, an iPSC model offers the greatest genetic insight into patients with monogenetic causes.

In 2011, the first iPSC line with genetically linked PD was established with an A53T mutation in the *SNCA* gene [74]. The mutation was subsequently fixed and both mutant and edited cell lines were co-differentiated to tyrosine hydroxylase positive (TH+) neurons. Dozens of lines have also been taken from members of the Iowa kindred with duplications and triplications of the *SNCA* gene [75]. These lines have shown increased sensitivity to neurotoxins and oxidative stressors, indicating a more accurate model of PD, but with healthy skepticism as the addition of toxins may not accurately portray the underlying mechanisms of PD [52]. Nevertheless, these models are useful in exploring affected patients’ endogenous response to environmental damage, possibly indicating mitochondrial malfunction. While this list is in no way exhaustive, genetic susceptibility has also been quantified in multiple iPSC lines with *LRRK2* [76,77,78], *PINK1* [79] and *GBA* mutations [80]. Perhaps iPSC technology will also be used as a diagnostic tool in the future to predict individual susceptibility to PD as well as a sourcing cell therapy.

The predisposition of DAn death in the ventral midbrain had long eluded models, but a new generation of iPSC mutant lines meets the challenge. Additionally, iPSC lines are established without the sacrifice of human zygotes or damaging side-effects in drug trials, allowing for the investigation of pathology without human harm or ethical concerns. These conditions are all foundational to two treatments only now within reach: cell therapy and patient-specific transplantation.

## 7. iPSCs as Cell Therapy for Parkinson’s Disease

Initial treatment of PD started in 1960 with the discovery that affected individuals lacked neurological dopamine. Clinicians began administering intravenous L-Dopa with almost immediate improvement of symptoms [81]. L-Dopa treatment remained the gold standard of treating PD for decades, but currently, there are other commercially available medications that target PD without a dopaminergic mechanism of action [82]. This increased focus on the dopaminergic pathway also helped in illuminating the motor circuitry of the basal ganglia by increasing funding to understand the broader interplay of PD’s pathophysiology. With a new understanding of the affected circuits, deep brain stimulation (DBS) by electrical stimulation to the internal segment of the globus pallidus and subthalamic nucleus was introduced as a supplemental therapy with dramatically positive results [83]. Time in the field of DBS has only improved the precision of electrode placement, higher flexibility and longer battery life to curb side-effects [83,84,85]. Like most neurological disorders, a number of studies have also shown the relative effectiveness of non-medical, non-surgical interventions such as exercise, dance and meditation [86,87].

Sparked by the genetic revolution, revealed subcellular mechanisms finally allow for a shift in focus from symptomatic therapies to the development of clinical-grade, disease-modifying treatment. Not for lack of interest, no disease-altering therapeutic options are currently available that can slow or halt the neurodegeneration of PD. However, there are a handful of drug candidates that show promise in varying stages of clinical trials for both general and sub-populations [88]. Such drugs include the LRRK2 inhibitor DNL201 in Phase I and the general dopamine agonist of dopaminergic D1 receptors PF-06412562 now in Phase III of clinical trials [36,89]. As noted above, clinicians are beginning to target the dysfunction of the specific biochemical pathways leading to PD using precision medicine.

The effectiveness of iPSCs has also been examined as a method to achieve such disease-altering treatment. Stem cells have already been used as therapy in a number of trials involving neural damage. The first occurred in 2010 with a clinical trial that used hESCs to treat spinal cord injury (SCI) [90]. Since then, stem cell-derived products have been used in other animal and human trials for therapy of neural disorders ranging from positive results with age-related macular degeneration (AMD) to poor results in AD [91,92]. These trials produced mixed results—in some cases highlighting the limitations of model organisms, due to cellular and gross anatomical differences, to predict the efficacy of treatment in human neurodegeneration [93].

With relatively localized neurodegeneration, PD is also a good candidate for cell therapy. Fetal tissue of the ventral midbrain was implanted in PD patients in 1987, and results from the trial showed cell survival and DAn functionality even 20 years after implantation in some cases [94,95]. As discussed above, the ethics of harvesting 4–10 embryos per patient and limitations to cell survival after preservation prevents fetal cell grafts as a viable form of cellular therapy on a national scale [96]. hESCs are also being utilized in a 2017 Chinese trial for PD, but it is too early to speculate on its effectiveness without conclusive data [97]. Though hESC implantation may produce promising results, strong immunosuppressants must be used to ensure that the graft is accepted. Safety in transplantation of all reprogrammed cells is paramount. Precautions must be taken to prevent infection, graft-induced dyskinesia and tumorigenesis when transplantation trials are conducted. Though cellular transplantation always poses some risk, the advantages of iPSCs present a viable future for cell therapy of PD. The history of these pertinent cell-based therapies can be visualized in Figure 2.

The first advantage is that iPSC lines can be established without the sacrifice of human embryos, removing a large ethical obstacle of human stem cell treatments. iPSCs also permit human leukocyte antigen (HLA) matches in patient-specific treatments, effectively reducing the severity of post-operational immunosuppressants. Histocompatibility has additionally shown a reduced immune response of lymphocytes and microglia as well as increased cell survival in iPSC transplantation of DAns in primate studies [98]. iPS dedifferentiation and reprogramming may be lengthy and burden the patient with high cost but reduced immune rejection and generic donor lines could significantly reduce costs when scaled up. The steps required for patient-specific transplantation of iPSCs are outlined in Figure 3. In Japan, researchers estimate that 50 iPS lines from HLA-homozygous donors will cover 73% of the Japanese population by matching three HLA loci (A, B and DR) [62]. Primate studies have already demonstrated significant improvements two years after transplanting human iPSCs into the non-human primate model of PD [99]. These human iPSC transplants into non-human primates functioned as midbrain DAns with improvement in scored movement. Similarly, postmortem analyses of iPSC transplants in non-human primate models reveal robust growth, proliferation and integration into existing neural networks [99,100]. Primate transplantation has also been achieved with clinical grade human ESCs and showed no significant tumor formation, as well as significant recovery of movement [101]. Clinical efficacy of treating PD with iPSC grafts in humans, however, will be established in the approaching future.

The primary difference between primate and human studies will be the treatment of actual PD patients as opposed to chemically-induced primate models. A team headed by Takahashi announced the first human clinical trial of iPSC-generated DAn transplantation to treat PD [102]. The trial began August 1st, 2018 at Kyoto University Hospital [57]. Cells will be sourced from third-party donors with matching HLA loci to ensure genetic integrity and eliminate the patients’ genetic interference. Additional testing was performed in rat models to confirm efficacy [103]. However, patients will still receive immunosuppressants due to the trial’s exploratory nature. Approximately 5 million cells will be administered to the SNpc through two drilled holes in the skull [104]. Seven patients with moderate PD have been selected for the trial with the benefit of cell therapy earlier on in neurodegeneration but also posing greater experimental risks and will be followed for two years. Clinicians will monitor the progression of the disease, as well as other side effects of too much dopamine in the SNpc that would result in involuntary movements. While many questions and ethical issues surrounding iPSCs and genetic engineering remain, the future for PD looks promising. However, iPSC treatment of PD will likely not completely restore function and should be examined within the context of other treatment options.

## 8. Conclusions

The loss of even a small cluster of 7800 DAns in the SNpc may result in severe debilitation in PD patients. Though cell death may arise from a number of postulated mechanisms, ultimately neuron survival is integral for proper motor and cognitive function. With no current therapies to recover from critical cell death, iPSCs provide an alternate route to potentially restore a disease-free state. Instead, patient-specific cells that are not predisposed to PD may be transplanted back into the SNpc, an ambition to restore function finally within reach. Programmed cell death does not afflict PD patients alone. However, iPSC therapies are better suited for PD as opposed to other forms of neurodegeneration because PD neuronal death is relatively localized. Cell therapies can, therefore, be targeted and less invasive than would be required to combat more diffuse and extensive degeneration such as ALS or AD. As has been previously shown in animal and primate models, patient-specific cell therapies, in combination with personalized pharmacogenetics, may offer the way forward to mitigating PD’s crippling form of neurodegeneration in humans.

## Figures and Tables

**Figure 1 cells-08-00026-f001:**
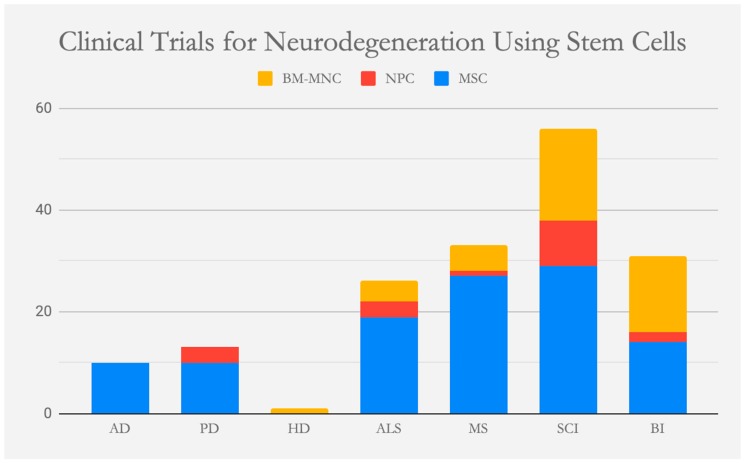
Clinical trials that have been or are being conducted worldwide to treat neurodegeneration using stem cells. To date, 170 clinical trials have employed mesenchymal stem cells (MSC), neural progenitor stem cells (NPC) and bone marrow-derived mononuclear cells (BM-MNC) in attempts to alleviate the neurodegeneration of Alzheimer’s disease (AD), Parkinson’s disease (PD), Huntington’s disease (HD), amyotrophic lateral sclerosis (ALS), multiple sclerosis (MS), spinal cord injury (SCI) and brain ischemia (BI). The data was collected from https://clinicaltrials.gov on 21 December 2018.

**Figure 2 cells-08-00026-f002:**
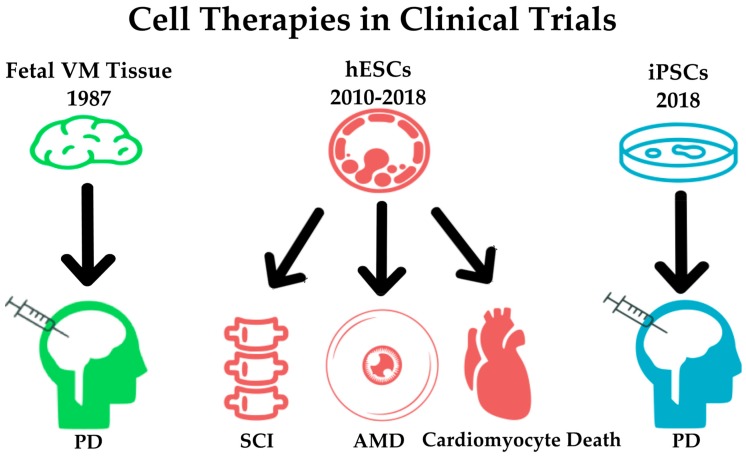
The progression of pluripotent cell-based therapies within the context of Parkinson’s Disease (PD) research. Beginning in 1987, fetal ventral midbrain (VM) tissue used as the cell source for the first clinical trial using cells to treat PD. In recent years, human embryonic stem cells (hESCs) are being utilized in a number of clinical trials involving neurodegeneration. Use of hESCs has shown special promise in spinal cord injury (SCI), age-related macular degeneration (AMD) as well as cell damage to the heart. In the summer of 2018, clinicians are beginning to undertake the first human trial using induced pluripotent stem cells (iPSCs) as a cell source to treat PD [57]. This trial will follow seven patients over the course of two years. The outcomes of these trials are detailed in the text.

**Figure 3 cells-08-00026-f003:**
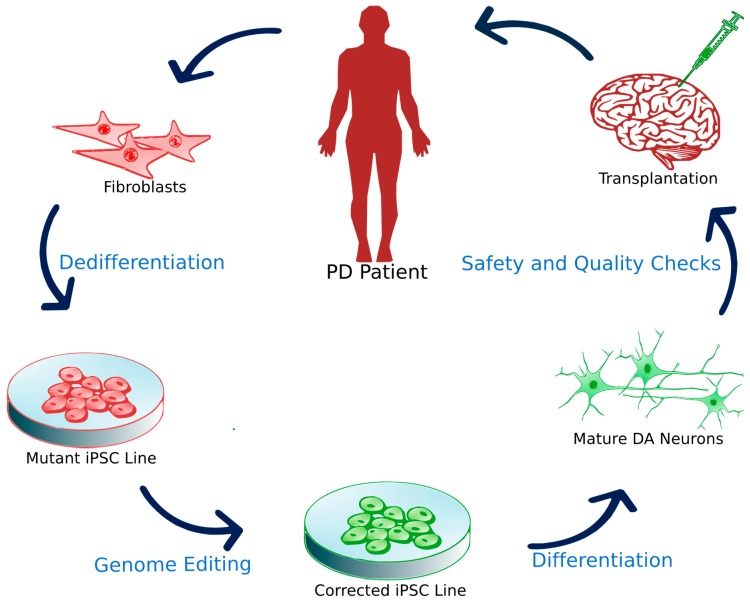
iPSC transplantation. First, fibroblasts are obtained from a patient afflicted with familial PD. Researchers express major reprogramming transcription factors to establish a mutant iPSC line. Using ZNF/TALEN or CRISPR/Cas9 technology, the significant mutation is corrected and then the line is differentiated into mature or progenitor DA neurons in xeno-free conditions. After sufficient quality assurance, the differentiated cells can then be used for clinical trials.

**Table 1 cells-08-00026-t001:** Major Familial Forms and Genetic Factors of Parkinson’s Disease.

PD Inheritance	Disease, (Mutations)	Gene, Location	Gene Function	Phenotype
Autosomal dominant	*PARK-SNCA*, (*A53T*, duplication, triplication)	*SNCA*, 4q22-1	The *SNCA* gene codes for the alpha-synuclein protein that is widely expressed in presynaptic terminals of neurons. Alpha-synuclein maintains the production of vesicles involved in neuronal communication. Alpha-synuclein is also thought to play a role in dopamine expression of voluntary and involuntary movement pathways.	Early-onset PD. Neurodegeneration within the SNpc and Lewy Body formation throughout the brain.
	*PARK-LRRK2*, (*G0219s*, *R1441C*)	*LRRK2*, 12q12	Encodes the leucine rich repeat kinase 2 protein, expressed in the cytoplasm and mitochondrial membranes of neurons. *LRRK2* is heavily involved in the ubiquitination of molecules, leading to their degradation. The precise function in PD is not known, but it is thought to coordinate neuronal survival and differentiation in the midbrain.	Late-onset PD with mixed neuropathology. Some cases present with Lewy Body formation and DAn death in the SN, others without Lewy Body formation.
Autosomal recessive	*PARK-DJ1*, (*Q456X*, *V170G*)	*DJ-1*, 1p36.23	Encodes the protein DJ-1, found in the brain and other tissues throughout the body. DJ-1 is a multi-functional protein with roles involved in the prevention of alpha-synuclein aggregation, neuronal protection under conditions of oxidative stress, transcriptional regulation and prevention of metal-induced cytotoxicity. All or some of these functions may be involved in some types of early PD formation.	Conclusive data has not been reported.
	*PARK-PINK1*, (exon 7 deletion)	*PINK1*, 1p36.12	Codes for the protein PTEN-induced putative kinase 1, located within mitochondria. *PINK1* exhibits a protective function of mitochondria during cellular stress by causing the parkin protein to bind to depolarized mitochondria and induce autophagy.	Early-onset PD complete with Lewy Body formation and acute DAn loss in the SNpc.
Genetic risk factor	Gaucher Disease (*L444P*, *N370S*)	*GBA*, 1q22	Codes for an enzyme active in lysosomes and cellular membranes. Beta-glucocerebrosidase is a housekeeping enzyme hydrolyzes the beta-glucosidic linkage of glucocerebroside into glucose and ceramide. Mutation causes glucocerebroside buildup in macrophages.	Severe neurological complications in addition to liver failure, bone lesions and low blood cell counts.

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
