# Peer review of "Treatment of Parkinson’s Disease through Personalized Medicine and Induced Pluripotent Stem Cells"

_cells, 2019, doi:10.3390/cells8010026_

Round 1

Reviewer 1 Report

The manuscript entitled “Treatment of Parkinson’s disease through personalized medicine and induced pluripotent stem cell” is a review article that emphasize light on the use of stem cells and personalized medicine (using genetics of PD) as a potential treatment for PD. The authors have given examples from many clinical studies and other translational studies to support their article. However the article is very weak in letting across the idea of what the authors really wanted to convey to the readers.

1.       There are a lot of information made available to the readers however there is no transitions between the different concepts and ideas. The authors are not able to convey how iPSCs and personalized medicine help PD. They are trying to jump from one idea to the other and it is very difficult to comprehend what the authors are trying to convey.

3.       More information in the abstract and introduction can be given. The current format/information is not very convincing. Please mention iPSCs in the introduction.

4.       Regarding the genetics aspects of the PD, the authors have given mutation analysis examples from various studies, but again, they are scattered all over the place in the manuscript.

5.       Many of the abbreviations needs to be deciphered.

6.       The authors should include more of in vivo study references.

7.       There are many strong statements that needs to be underplayed – for example, the last few sentences in the abstract are very strong.

8.       Figure 2 – Be consistent, between the Fetal tissue and iPSCs, use arrows for both, or no arrow.

Overall, this manuscript needs to be re-written in a cohesive manner putting all the ideas together and conveying to the readers. Because this topic is already very well documented and reviewed in the current literature, this manuscript, in this format, does not add anything to the current knowledge

This reviewer rejects this manuscript in the current format. 

Author Response

1. There are a lot of information made available to the readers however there is no transitions between the different concepts and ideas. The authors are not able to convey how iPSCs and personalized medicine help PD.

Heavily edited the abstract to better convey the advantages of iPSC models.

Now reads: 

Parkinson’s Disease (PD) is an intractable disease resulting in localized neurodegeneration of dopaminergic neurons of the substantia nigra pars compacta. Many current therapies of PD can only address the symptoms and not the underlying neurodegeneration of PD. To better understand the pathophysiological condition, researchers continue to seek models that mirror PD’s phenotypic manifestations as closely as possible. Recent advances in the field of cellular reprogramming and personalized medicine now allow for previously unattainable cell therapies and patient-specific modeling of PD using induced pluripotent stem cells (iPSCs). iPSCs can be selectively differentiated into a dopaminergic neuron fate naturally susceptible to neurodegeneration. In iPSC models, unlike other artificially-induced models, endogenous cellular machinery and transcriptional feedback are preserved, a fundamental step in accurately modeling this genetically complex disease. In addition to accurately modeling PD, iPSC lines can also be established with specific genetic risk factors to assess genetic sub-populations’ differing response to treatment. iPS cell lines can then be genetically corrected and subsequently transplanted back into the patient in hopes of re-establishing function. Current techniques focus on iPSCs because they are patient-specific, thereby reducing the risk of immune rejection. The year 2018 marked history as the first human trial for PD iPSC transplantation began in Japan. This form of cell therapy has shown promising results in other model organisms and is currently one of our best options in slowing or even halting the progression of PD. Here we examine the genetic contributions that have reshaped our understanding of PD, as well as the advantages and applications of iPSCs for modeling disease and personalized therapies.

2. They are trying to jump from one idea to the other and it is very difficult to comprehend what the authors are trying to convey.

Moved pluripotent stem cell and genetic paragraphs to have a more logical flow.

3.       More information in the abstract and introduction can be given. The current format/information is not very convincing. Please mention iPSCs in the introduction.

The abstract was heavily edited. If you feel other changes are needed, please let the authors know.

4.       Regarding the genetics aspects of the PD, the authors have given mutation analysis examples from various studies, but again, they are scattered all over the place in the manuscript.

Changed the format to introduce pluripotent stem cells after the sections on genetics and personalized medicine.

5.       Many of the abbreviations needs to be deciphered.

The authors included descriptors and names for all of the genes including LRRK2, SNCA, PINK1, DJ1, DRD2, DAT1, GBA. If additional abbreviations should be clarified please notify the authors.

6.       The authors should include more of in vivo study references.

Included 3 more promising primate studies with iPSC and ESC transplantation within the last 2 years.

7.       There are many strong statements that needs to be underplayed – for example, the last few sentences in the abstract are very strong.

The limited application of cell therapies to broader, diffuse neurodegeneration (like Alzheimer’s) was made more distinct. If other claims seem too extreme, the authors would be happy to change them.

The last few sentences now read: Programmed cell death does not afflict PD patients alone. However, iPSC therapies are better suited for PD as opposed to other forms of neurodegeneration because PD neuronal death is relatively localized. Cell therapies can, therefore, be targeted and less invasive than would be required to combat more diffuse and extensive degeneration such as ALS or AD. As has been previously shown in animal and primate models, patient-specific cell therapies, in combination with personalized pharmacogenetics, may offer the way forward to mitigating PD’s crippling form of neurodegeneration in humans.

8.       Figure 2 – Be consistent, between the Fetal tissue and iPSCs, use arrows for both, or no arrow.

Included arrows for both.

Also included various grammatical corrections as well as other formatting and technical issues presented by the other reviewers.

Reviewer 2 Report

The review adequately covers important aspects of current and potential future therapeutic applications of iPSC to animal models of PD and human PD patients. The review manuscript was well written and the information timely.

Listed below are several comments that are organized to the sentence numbers in the review manuscript. The comments can be straightforwardly addressed and should improve the manuscript.

Line 36 A table listing neurodegenerative diseases and therapeutic SC applications, and their outcomes would be valuable.

Line 42 There are in practice therapies to diminish the progress of certain ng diseases. Perhaps include the adjective “novel” before therapies.

Uninitiated readers will appreciate a spelled-out name for the various genes. For example, SNCA encodes for the protein called alpha-synuclein. (also see GBA, Lrrk2, DRD2, etc…).

As well, a brief description of the function of the various proteins mentioned would improve the review (as has been provided for alpha-synuclein addressed on line 113).

Finally, it would be helpful to provide the information at the first mention of the protein in the text of the review, i.e., move sentences 113/114 to line 53.

Clarify which model is noted in the sentence Line 67.

Line 73 indicate which models.

Line 74 …“when executed with optimal methods of action”…  (unclear).

Line 88 If available, provide a URL to the clinical trial. Is it the same trial mentioned in the legend to Fig. 1, and the closing comments of the review (Line 350)?

Line 108 The information implies the 5 different missense mutations are in the SNCA gene. Please clarify.

Line 256 …co-cultures were both differentiated… The statement is unclear, perhaps rewrite.

Author Response

Point 1: Line 36 A table listing neurodegenerative diseases and therapeutic SC applications, and their outcomes would be valuable.

I included a figure to better visualize the data

Point 2: Line 42 There are in practice therapies to diminish the progress of certain ng diseases. Perhaps include the adjective “novel” before therapies.

Included novel 

Point 3 and 4: Uninitiated readers will appreciate a spelled-out name for the various genes. For example, SNCA encodes for the protein called alpha-synuclein. (also see GBA, Lrrk2, DRD2, etc…).

As well, a brief description of the function of the various proteins mentioned would improve the review (as has been provided for alpha-synuclein addressed on line 113).

The authors included descriptors and names for LRRK2, SNCA, PINK1, DJ1, DRD2, DAT1, GBA. 

Point 5: Finally, it would be helpful to provide the information at the first mention of the protein in the text of the review, i.e., move sentences 113/114 to line 53.

Also moved line 110 to line 53

Point 6: Clarify which model is noted in the sentence Line 67.

Included “artificially induced model”

Point 7: Line 73 indicate which models.

Changed to mouse models 

Point 8: Line 74 …“when executed with optimal methods of action”…  (unclear).

Removed “when executed with optimal methods of action”

Point 9: Line 88 If available, provide a URL to the clinical trial. Is it the same trial mentioned in the legend to Fig. 1, and the closing comments of the review (Line 350)?

Included and also referenced in Fig. 1 and closing comments

Point 10: Line 108 The information implies the 5 different missense mutations are in the SNCA gene. Please clarify.

Now reads: five other missense mutations to the SNCA gene,

Point 11: Line 256 …co-cultures were both differentiated… The statement is unclear, perhaps rewrite.

Now reads: The mutation was subsequently fixed and both mutant and edited cell lines were co-differentiated to tyrosine hydroxylase positive (TH+) neurons

Also included various grammatical corrections as well as other formatting and technical issues presented by the other reviewers.

Reviewer 3 Report

This review covers a number of elements directly related to the treatment of Parinson's disease with an eventual focus of iPSC treatments.  With some minor revisions, the review will provide useful information for the general scientific community.

10 The Basalis nucleus of Meynert is primarily cholinergic.  Also, should use a more updated  reference: Liu et al., Acta Neuropathol. 129:527, 2015.

Page 2 line 65 - 6-hydroxydopamine is not given intraperitoneally, it is administered directly into the CNS

Page 3 line 88 - provide the clintrial number

Page 3 line 118 - This sentence is awkward - restructure

Page 4 - line 128 - symptoms associated with remove to

Page 6 line 170 - reference clintrials number

line 186 - patients'

page 7 - line 193  replace can with should

line 204 - replace rewind with reprogram

Page 9 - line 283 - clarify this sentence

lines 287-293 - You should elaborate on the handful of drug candidates, since this is a major theme in the review

Conclusion - A distinction should be made between the application of iPSCs for Parkinson's and more diffuse neurological disorders such as Alzheimers and ALS where the area of treatment is far more expansive in comparison to Parkinson's.

Author Response

Point 1: The Basalis nucleus of Meynert is primarily cholinergic.  Also, should use a more updated reference: Liu et al., Acta Neuropathol. 129:527, 2015.

The reference was updated and we included "primarily cholinergic" before the BNoM. 

Point 2: line 65 - 6-hydroxydopamine is not given intraperitoneally, it is administered directly into the CNS

modified to "direct injections into the central nervous system" 

Point 3: line 88 - provide the clintrial number

Included clinical trial number UMIN000033564 and reference to Japanese clinical trial database

Point 4: line 118 - This sentence is awkward - restructure

Changed to "In 2002, Funayma et al. reported that a region of chromosome 12 was found to be linked to PD inheritance in a Japanese family."

Point 5: line 128 - symptoms associated with remove to

now reads: Curiously, patients with LRRK2 PD experience less severe motor symptoms associated with the frequency of falls and progression of dyskinesia

Point 6: Page 6 line 170 - reference clintrials number

Included ID NCT03710707

Point 7: line 186 -   patients' 

Fixed

Point 8page 7 - line 193  replace can with should

Replaced

Point 9: line 204 - replace rewind with reprogram

Replaced

Point 10: line 283 - clarify this sentence 

The numbering system was off so I assumed you meant this sentence. "This increased focus on the dopaminergic pathway also helped in illuminating the motor circuitry of the basal ganglia by increasing funding to understand the broader interplay of PD's pathophysiology." 

Point 11: lines 287-293 - You should elaborate on the handful of drug candidates, since this is a major theme in the review

 Included the sentence with references "Such drugs include the LRRK2 inhibitor DNL201 in Phase I and the general dopamine agonist of dopaminergic D1 receptors PF-06412562 now in Phase III of clinical trials."

Point 12: Conclusion - A distinction should be made between the application of iPSCs for Parkinson's and more diffuse neurological disorders such as Alzheimers and ALS where the area of treatment is far more expansive in comparison to Parkinson's

Now reads: Programmed cell death does not afflict PD patients alone. iPSC therapies for PD, however, possesses substantial promise over cell therapies fighting other forms of neurodegeneration because PD neuronal death is relatively localized. Cell therapies can, therefore, be targeted and less invasive than would be required to combat the more diffuse and extensive degeneration of ALS or AD. As has been previously shown in animal and primate models, patient-specific cell therapies, in combination with personalized pharmacogenetics, may offer the way forward to mitigating PD’s crippling form of neurodegeneration in humans.

Also included various grammatical corrections as well as other formatting and technical issues presented by the other reviewers.

Round 2

Reviewer 1 Report

The authors have rewritten the abstract which now aligned better with the text presented. Reorganization of the sections is helpful too. Addition of Figure 1 on clinical trials is great. Fig 2, previously Fig 1, was not changed on the r1 document, the arrow for fetal cells is still missing and the font of text needs to be changed.

Thank you for editing the manuscript.

Author Response

- the arrow for fetal cells is still missing and the font of text needs to be changed.

Figure 2 was modified to include an arrow for the fetal cells and the text was changed to Palatino Linotype.